# Environmental Concern, Environmental Knowledge, and Residents' Water Conservation Behavior: Evidence from China

**Yong Li** [1], **Bairong Wang** [2,*] **and Manfei Cui** [3]

[1] School of Marxism, Shanghai Maritime University, Shanghai 201306, China; liyong@shmtu.edu.cn
[2] School of Economics and Management, Shanghai Maritime University, Shanghai 201306, China
[3] School of Public Affairs, Zhejiang University, Hangzhou 310058, China; 11922050@zju.edu.cn
[*] Correspondence: bairongw@buffalo.edu

**Abstract:** Water conservation represents a typical green behavior and a sustainable lifestyle. Understanding residents' water conservation behaviors is a prerequisite for promoting more water savers. Using the snowball sampling technique, this study conducted a survey on a sample of 532 Chinese residents to investigate their water conservation behavior, i.e., reusing water in daily life. This study aims for examining psychological and knowledge factors on residents' water conservation behavior in China using binary logistic regression. Results show that over half of the respondents (54%) have the habit of reusing water in their daily lives. Residents with stronger environmental concern and higher level of environmental knowledge are more likely to exhibit household water conservation in China. Additionally, environmental knowledge plays a positive moderating role in the relationship between environmental concern and water conservation behavior. Environmental knowledge serves as a catalyzer that facilitates the transformation from residents' environmental concern into real water conservation behavior. Among the demographic variables, only income exerts significantly negative effect on residents' water conservation behavior, and other variables (e.g., age and gender) fail to exert any influence on this behavior. This study contributes to the literature on environmental psychology and concludes with implications for water resource management.

**Keywords:** environmental concern; environmental knowledge; water conservation; moderating effect; China

---

## 1. Introduction

Increasing environmental problems have raised wide concerns around the world. To tackle the environmental problems, one effective solution is to promote environmentally friendly and sustainable development. One important emphasis of sustainable development is to use natural resources reasonably given the increasing speed of resources depletion [1–3]. For instance, it is reported that the world is seeing an increase in demand for fresh water resources due to rising population and extreme weather caused by climate change [4]. By 2030, the world is expected to face a 40% fresh water shortage [5]. The scarcity of fresh water has been identified as one of the main environmental challenges [6–8]. To solve this problem, the sustainable use of water from the individual level is an effective method [4,9,10]. Water conservation is regarded as a typical pro-environmental behavior [6]. Many efforts have been made to study the influential factors of multiple pro-environmental activities [1,11,12], such as reusing plastic bags [13], green purchase behavior [14,15], energy savings [16], and energy sustainability policy design [17]. However, little research has been conducted regarding determinants of household water conservation behavior, especially in developing countries [18]. With the largest population, rising water demand, and water scarcity, China is facing severe water crisis [19]. Thus, this study is motivated to fill the research gap by investigating the determinants of residents' water conservation behavior in China.



Previous studies suggest that except for socio-demographic variables, whether people choose to conduct green behaviors is mainly affected by psychology and knowledge factors [20,21]. Regarding the psychological drivers of pro-environmental behaviors, existing studies indicate that environmental concern is a powerful one [6,22,23]. Moreover, some literature points out that people's environmental knowledge is necessary to facilitate the transformation from their environmental concern into real environment-protecting behavior [6,24]. Water conservation behavior can only be promoted if the key influential factors of such behavior are identified and understood [25]. Thus, we are motivated to examine how environmental concern and environmental knowledge jointly impact water conservation behavior in China.

Contribution of this research is three-fold. First, this study contributes to uncover the status quo of household water conservation in China by conducting a semi-structured survey. Second, with the techniques of binary logistic regression, this study identifies a direct effect of environmental concern on water conservation behavior in China. Third, this study explores the indirect effect of environmental knowledge on water conservation and finds a moderating role of environmental knowledge in transforming environmental concern into real water conservation behavior in China.

The rest of this study is constructed as follows: Section 2 describes theoretical background and research hypotheses; Section 3 presents materials and methods; Sections 4 and 5 present and discuss the analysis results, respectively; Section 6 focuses on limitations and future research.

## 2. Theoretical Background and Research Hypotheses

The purpose of this study is to explore the influential factors of Chinese residents' water conservation behaviors. Existing studies have discussed the psychological and knowledge-related influential factors of pro-environmental attitudes and behaviors [6,12]. For instance, based on the Theory of Planned Behavior proposed by Ajzen [26], existing studies found a positive relationship between people's subjective norms and water conservation behavior [27–29]. Among the various potential influential factors, environmental concern serves as one of the important starting points for pro-environmental behaviors. When people start to show their concerns and care for the environment, they are more likely to form their intention or plan for pro-environmental behaviors. However, even with enough passion for environmental protection, accurate knowledge or method is also necessary. Moreover, existing studies also show that environmental concern [25] and environmental knowledge [6] are effective drivers to residents' water conservation behavior. In this regard, this study selects environmental concern and environmental knowledge as the investigating variables for analysis.

### 2.1. Environmental Concern and Pro-Environmental Behavior

Environmental concern refers to the degree to which people are aware of environmental problems and their willingness to solve these problems [30]. As mentioned by Wang and Li [12], environmental concern exhibits an individual's strong attitude towards protecting the environment. Previous research has highlighted the importance and significance of environmental concern in explaining individuals' pro-environmental behaviors [31–34]. For example, environmental concern is found to have a direct and positive impact on gas emission [35], garbage reduction [23], and recycling and conservation [36]. Similarly, Pagiaslis and Krontalis [24] suggest that individuals with high environmental concern are more willing to pay extra for environmentally friendly products. As for water conservation behavior, Aprile and Fiorillo [6] argue that there is a positive link between environmental concern and water conservation in Italy. A study in Australia shows that people are more likely to conserve water when they care more about the environment and the limited water resources [25]. Therefore, we assume that residents' environmental concern will positively impact household water conservation behavior in China and propose the following Hypothesis 1:

**Hypothesis 1 (H1).** *Residents' environmental concern has a significantly positive effect on water conservation behavior.*

*2.2. Environmental Knowledge and Pro-Environmental Behavior*

Environmental knowledge indicates individuals' cognition of environmental issues and "general knowledge of facts, concepts, and relationships regarding the natural environment and its major ecosystems" (Fryxell and Lo, 2003). Generally, an individual is unlikely to care about the environment or conduct pro-environmental behavior if he/she knows little about the environment problem [24,37,38]. For instance, Kennedy, Beckley, McFarlane and Nadeau [39] find that more than 60% of their respondents feel that their pro-environmental behavior is often constrained by a lack of environmental knowledge in Canada. Moreover, it is difficult to make wise environmental choices if an individual has inaccurate environmental knowledge [40,41]. Environmental knowledge has been demonstrated as one of the most potent predictors of pro-environmental behaviors [40,42–45], such as green food purchase [46] and green tourism behavior [47]. Likewise, Polonsky et al. (2012) point out that consumers are more likely to choose pro-environmental products when they obtain more information about them. As water conservation is a typical pro-environmental behavior [6], this study proposes that environmental knowledge has a positive influence on residents' water conservation behavior in China. The following Hypothesis 2 is proposed:

**Hypothesis 2 (H2).** *Residents' environmental knowledge has a significantly positive effect on water conservation behavior.*

*2.3. The Interacting Effect of Environmental Concern and Environmental Knowledge*

It has been suggested that environmental knowledge is a necessary but not sufficient pre-condition for pro-environmental behavior [48,49]. Environmental knowledge may interact with other influential factors to impact pro-environmental behavior. Similarly, Varela-Candamio, Novo-Corti and García-Álvarez [50] pointed out that the effect of environmental knowledge on pro-environmental behavior could be indirect.

Milfont and Schultz [51] argued that the majority of the world's population has expressed environmental concern. However, in terms of green actions, the results are far from satisfying [13]. "All talk and no action" is a puzzle in explaining the green behaviors. Hence, different green behaviors of individuals with the same levels of environmental concern could be analyzed in combination with environmental knowledge. Specifically, individuals who have insufficient environmental knowledge and unfamiliar with the approaches of environmental protection are unlikely to engage in green actions. This "information gap" is a barrier to pro-environmental behaviors [52] even though people have environmental concerns. For instance, Dolnicar and Hurlimann [10] found that Australians' high concern for water conservation is not always translated into water conservation action. Thus, we assume that environmental knowledge may interact with environmental concern to influence residents' water conservation behavior in China. In this view, this study proposes the following Hypotheses 3a and 3b:

**Hypothesis 3a (H3a).** *Environmental knowledge moderates the relationship between environmental concern and water conservation behavior.*

**Hypothesis 3b (H3b).** *Environmental concern moderates the relationship between environmental knowledge and water conservation behavior.*

**3. Materials and Methods**

This study conducted an online semi-structured survey from November to December 2021 to investigate resident's water conservation behavior in China. The questionnaire consists of three sections. The first section deals with the demographic information of the sample, and the investigated variables include age, gender, and education level, etc. The second section deals with the environment-related variables, e.g., environmental concern

and environmental knowledge. The third section investigates residents' water conservation behavior and other pro-environmental behaviors. We distributed questionnaire surveys online to the general public in China using the snowballing technique. Utilizing existing social and personal contacts, respondents were recruited through WeChat, a popular social media platform in China. Before widely distributing the questionnaire, a pilot study was completed with 25 respondents to confirm the statements are accurate and understandable. Finally, the study successfully distributed a total of 534 questionnaires, among which 532 were valid for analysis. According to the rules proposed by Tharenou, Donohue, and Cooper [53], for each variable in the model, at least 25 effective responses are required. In this study, a total of 8 variables are analyzed, and therefore, at least 200 effective respondents are necessary. The sample size is large enough in this study. In addition, the respondents are from 25 out of 31 provinces in China's mainland.

The dependent variable, i.e., water conservation behavior, was measured by the question "Do you have the habit of reusing water in your daily life, such as keeping the water for washing vegetables to flush the toilet?" In this study, water conservation specifically refers to household water reuse. Responses were coded into a binary variable: 1 = "yes", and 0 = "no". Regarding the independent variables, environmental concern is measured by items adapted from Minton and Rose [54]. We chose 9 questions among the original 16 questions from Minton and Rose [54]. To make the survey as concise as possible, we removed repeated similar questions and those not suitable for Chinese culture and situation. The sampling items of environmental concern are "I think the government should devote more money toward supporting conservation and environmental programs; and environmental issues are overrated and do not concern me (Reversed)". All items were measured on a five-point Likert scale, with 1 = "completely disagree", and 5 = "completely agree". The Cronbach's $\alpha$ of environmental concern is 0.887, which is reliable. Regarding environmental knowledge, based on the measurement proposed by Aprile and Fiorillo [6], the variable in this study was measured by the question "How often do you get access to environmental knowledge, such as watching documentaries, TV programs, or short videos related to environmental protection?". This item was measured on a five-point Likert scale, with 1 = "never", and 5 = "always". Demographic information includes age, gender, education, marital status, and monthly income. For instance, Gilg and Barr [55] suggested that elderly people were more likely to be water savers. In addition, existing studies indicated that higher income was associated with heavier water consumption [2,3,56]. As water conservation is a dummy variable, binary logistic model was used to conduct data analysis in this study.

A statistical summary of the sample is presented in Figure 1. In general, the respondents are relatively young and well-educated, as over 80% of them were younger than 40 and with at least a bachelor's degree. Specifically, 47.2% of the respondents were less than 30 years old, 34.8% were 31–40 years old, 12.6% were 41–50 years old, and the other 5.5% were over 50 years old. As for gender, 55.6% of the survey respondents were female, and 44.4% were male. In terms of education level, 85.4% of the respondents had a bachelor's degree or higher. The marital status of the respondents was distributed evenly, as 52.6% were married, and 47.4% were single. Regarding the distribution of income, 32.3% of the respondents earned < RMB 5000 monthly, 37.2% earned RMB 5000–9999 monthly, 22.9% earned RMB 10,000–19,999 monthly, and the other 7.5% earned ≥ RMB 20,000 monthly. Among the 532 respondents, 53.6% of them had the habit of conserving water, while the remaining 46.4% did not. About half of the investigated respondents were not water savers, which was far from satisfying. This research finding indicates that there is significant room to increase Chinese residents' household water conservation behavior in the future.

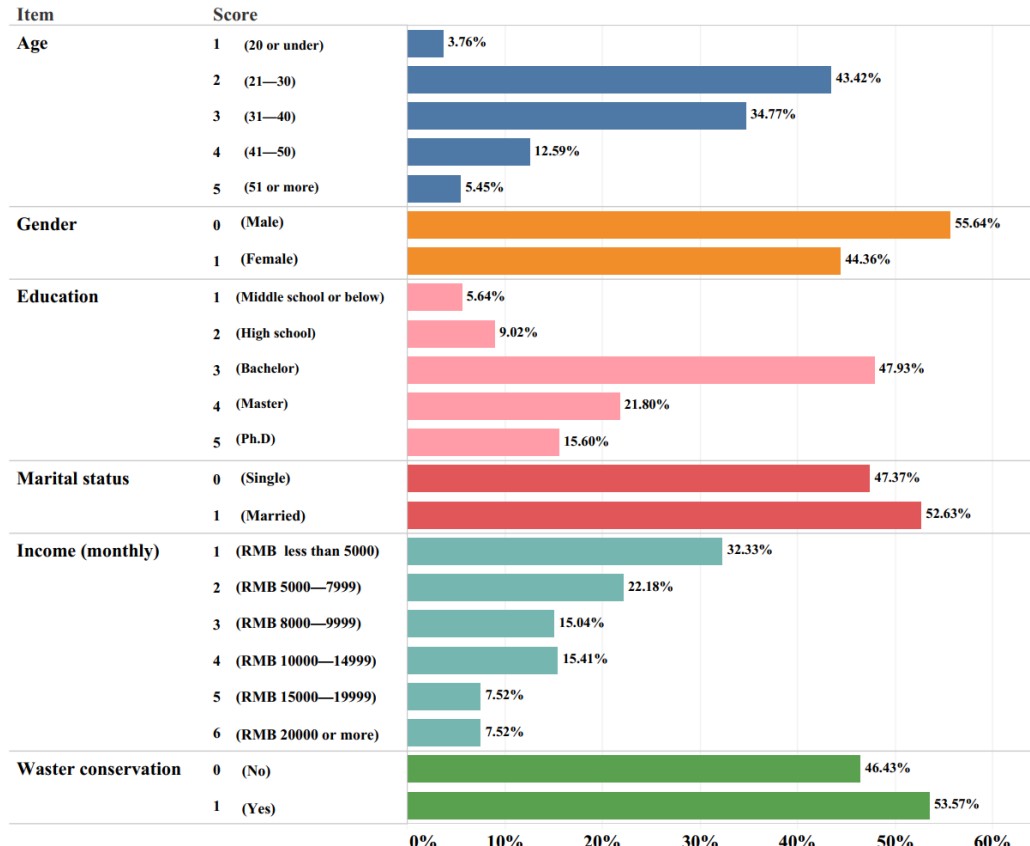

**Figure 1.** Distribution of the sample (N = 532).

## 4. Analysis and Results

Table 1 shows the means, standard deviations, and correlation coefficients of the main variables in this study. The mean of water conservation behavior is 0.54; namely, about half of the respondents have developed a water conservation behavior by the time of this study. The mean of environmental concern is 4.12, suggesting that most of the investigated respondents have high environmental concern, in line with the findings of Milfont and Schultz [51]. In addition, the mean of environmental knowledge is 2.87, indicating a lack of environmental knowledge among the respondents.

**Table 1.** Means, standard deviations, and correlation coefficients of the model variables (N = 532).

| Variable | Mean | SD | 1 | 2 | 3 | 4 | 5 | 6 | 7 | 8 |
|---|---|---|---|---|---|---|---|---|---|---|
| 1. Age | 2.73 | 0.92 | 1 | | | | | | | |
| 2. Gender | 0.44 | 0.50 | 0.065 | 1 | | | | | | |
| 3. Education | 3.33 | 1.03 | −0.247 ** | 0.062 | 1 | | | | | |
| 4. Marital | 0.53 | 0.50 | 0.513 ** | −0.039 | −0.116 ** | 1 | | | | |
| 5. Income | 2.66 | 1.59 | 0.147 ** | 0.135 ** | 0.276 ** | 0.198 ** | 1 | | | |
| 6. Environmental concern | 4.12 | 0.64 | 0.003 | −0.123 ** | 0.033 | −0.011 | 0.072 | 1 | | |
| 7. Environmental knowledge | 2.87 | 0.92 | 0.006 | 0.013 | −0.010 | −0.017 | −0.028 | 0.164 ** | 1 | |
| 8. Water conservation behavior | 0.54 | 0.50 | 0.021 | −0.018 | −0.115 ** | −0.023 | −0.139 ** | 0.105 * | 0.170 ** | 1 |

Notes: * $p < 0.05$, ** $p < 0.01$.

Table 2 summarizes the effects of environmental concern, environmental knowledge, and demographic variables on residents' water conservation behavior. As shown in Model 4, income has a significantly negative effect ($\beta = -0.158$, $p < 0.05$) on residents' water conservation behavior. Compared with the high-income group, the low-income group is more prone to exhibit water conservation. This result is similar to existing findings, which show that higher income was associated with higher water consumption [2,3,56]. A possible reason behind this finding is that low-income residents have stronger economic motives to be water savers compared with their high-income counterparts. Therefore, to achieve better water conservation results, more economic incentives could be effective. However, other demographic variables, such as age, gender, education, and marital status show no significant influence on residents' water conservation behavior (see Model 4). The failure of age's influence on water conservation behavior is different from existing findings, which suggest that elderly people are more likely to be water savers [6,55]. One possible explanation of this finding is that water is not expensive in China, and fresh water is even free in some rural areas. Therefore, no matter old or young, married or single, female or male, as long as people are not economically motivated, they show no significant difference in water conservation behaviors. That is to say, water conservation behaviors are highly economically motivated in China.

As shown in Model 4 of Table 2, environmental concern positively impacts residents' water conservation behavior ($\beta = 0.371$, $p < 0.05$), in line with the findings of Tam and Chan [32] and Rhead, Elliot [33]. Thus, H1 is supported. Strong environmental concern is related to a higher likelihood of being a water saver. Moreover, consistent with the studies of Gifford and Nilsson [38] and Ouz and Kavas [37], environmental knowledge exerts a significantly positive effect ($\beta = 0.333$, $p < 0.01$) on residents' water conservation behavior (see Model 4). Having more knowledge of environmental protection is related to a higher likelihood of being a water saver. Therefore, H2 is supported. These results suggest that environmental concern and environmental knowledge are both crucial influential factors in predicting residents' water conservation behavior in China. As highlighted in this study, a lack of environmental concern or poor environmental knowledge may be a main barrier to the development of water conservation behavior. In addition, there is a significantly interacting effect between environmental concern and environmental knowledge ($\beta = 0.236$, $p < 0.05$) on residents' water conservation behavior (see Model 4). Hence, H3a and H3b are supported. Environmental knowledge has been examined as a moderator in existing environmental research [57–61]. By reviewing the previous studies, this study suggests that it is more plausible that environmental knowledge moderates the relationship between environmental concern and water conservation behavior.

To more clearly characterize the moderation mechanism, simple slope tests (see Figure 2) were conducted to evaluate whether the relationship (slope) between environmental concern and the likelihood of conserving water is intensified or weakened by different levels of environmental knowledge. As shown in Figure 2, environmental knowledge positively moderates the relationship between environmental concern and water conservation. To be specific, when environmental knowledge is high, environmental concern has a stronger positive influence on residents' water conservation behavior. However, when environmental knowledge is low, environmental concern has a weaker positive influence on individuals' water conservation behavior. Thus, this finding confirms that environmental knowledge serves as a catalyzer that facilitates the transformation from environmental concern into water conservation behavior. Based on this finding, the government is suggested to promote residents' environmental concern and environmental knowledge simultaneously in the future to stimulate more water conservation behaviors.

**Table 2.** Binary logistic regression results for water conservation behavior.

| Variable | | Water Conservation Behavior | | | | | | | | | | | |
|---|---|---|---|---|---|---|---|---|---|---|---|---|---|
| | | **Model 1** | | | **Model 2** | | | **Model 3** | | | **Model 4** | | |
| | | **B** | **Wald** | **Exp(B)** | **B** | **Wald** | **Exp(B)** | **B** | **Wald** | **Exp(B)** | **B** | **Wald** | **Exp(B)** |
| Control variables | Constant | 0.917 | 3.991 | 2.501 | −0.653 | 0.787 | 0.521 | −1.312 | 2.900 * | 0.269 | −1.598 | 4.085 ** | 0.202 |
| | Age | 0.074 | 0.402 | 1.077 | 0.069 | 0.141 | 1.071 | 0.061 | 0.263 | 1.063 | 0.075 | 0.392 | 1.078 |
| | Gender | −0.002 | 0.000 | 0.998 | 0.069 | 2.936 | 1.071 | 0.049 | 0.070 | 1.050 | 0.019 | 0.011 | 1.020 |
| | Education | −0.156 | 2.729 * | 0.856 | −0.163 | 2.936 * | 0.850 | −0.166 | 2.988 * | 0.847 | −0.150 | 2.426 | 0.860 |
| | Marital | −0.104 | 0.245 | 0.902 | −0.083 | 0.155 | 0.920 | −0.073 | 0.118 | 0.929 | −0.107 | 0.246 | 0.899 |
| | Income | −0.150 | 6.155 ** | 0.861 | −0166 | 7.386 *** | 0.847 | −0.161 | 6.808 *** | 0.851 | −0.158 | 6.489 ** | 0.854 |
| Independent variable | Environmental concern | | | | 0.391 | 7.392 *** | 1.478 | 0.314 | 4.614 ** | 1.369 | 0.371 | 6.030 ** | 1.449 |
| | Environmental knowledge | | | | | | | 0.350 | 11.802 *** | 1.419 | 0.333 | 10.138 *** | 1.395 |
| Interaction term | Environmental concern × Environmental knowledge | | | | | | | | | | 0.236 | 6.648 ** | 1.266 |
| −2 Log likelihood | | 720.622 | | | 713.072 | | | 700.835 | | | 693.794 | | |
| Chi-square test | | 14.170 ** | | | 21.720 *** | | | 33.957 *** | | | 40.998 *** | | |
| Cox and Snell $R^2$ | | 0.026 | | | 0.040 | | | 0.062 | | | 0.074 | | |
| Nagelkerke $R^2$ | | 0.035 | | | 0.053 | | | 0.083 | | | 0.099 | | |
| N | | 532 | | | 532 | | | 532 | | | 532 | | |

Notes: The significance of differences on variables is tested by Wald tests. N = Number of observations; *** $p < 0.01$, ** $p < 0.05$, * $p < 0.1$.

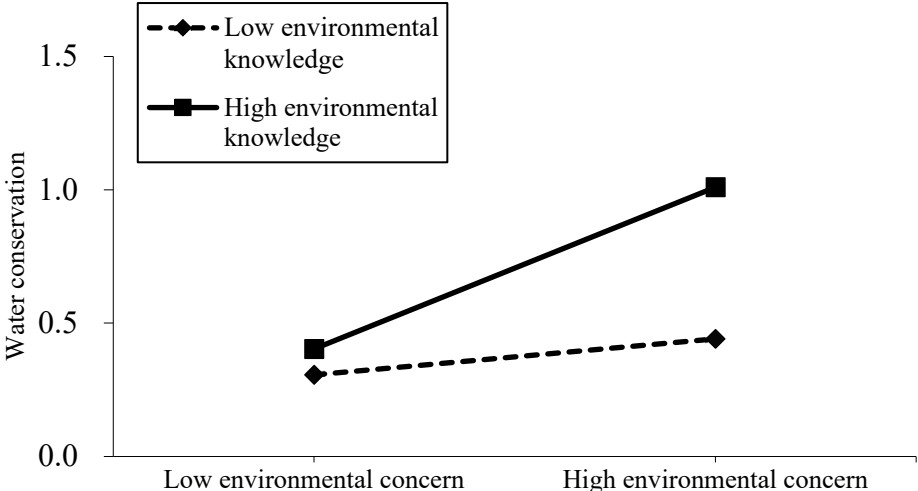

**Figure 2.** The moderating effect of environmental knowledge on the relationship between environmental concern and water conservation behavior.

## 5. Discussion

Water conservation has long been seen as a crucial aspect of water resource management [10]. This study conducts an online survey on a sample of 532 Chinese respondents to learn their water conservation behavior. The main contribution of this study is to examine the psychology and knowledge factors that may explain water conservation behavior through binary logistic regression. The research findings suggest that both environmental concern and environmental knowledge are vital drivers for residents' water conservation behavior in China, and environmental knowledge plays a moderating role on the relationship between environmental concern and water conservation. This study contributes to the literature on environmental psychology and concludes with implications for water resource management as follows:

First of all, water conservation represents a critical pro-environmental activity and a sustainable lifestyle. This study provides the status quo of water conservation behaviors in China, which definitely would contribute to the global water resource management and sustainable development. Based on the survey results, 46.4% of the investigated respondents do not have the habit of reusing water (see Figure 1), indicating that there is still significant room to increase water conservation in China. This finding highlights the necessity of promoting more water conservation behaviors and green lifestyle in the future. Furthermore, water conservation can be increased through water-saving technical improvements [5] and water-saving publicity campaigns [62].

Second, environmental concern is examined to be a significant driver of water conservation behavior, consistent with the findings of Western research, which shows that greater environmental concern is related to more engagement in green actions [31–33]. This study provides new evidence to confirm the positive effect of environmental concern on residents' water conservation in China. Individuals who have a higher level of environmental concern may be more sensitive to water shortage and are more likely to exhibit water conservation in daily life. Thus, based on the analysis, this study suggests that the cultivation of environmental concern regarding water scarcity is a potentially effective way. Moreover, as for the individuals with a low level of environmental concern, measures such as economic incentives or penalties are necessary to modify their water conservation behaviors. Water conservation is a common concern of the whole society. To pursue better water management results, social participation is necessary and important [63]; i.e., government, enterprise, and the public can all be involved in water conservation. For instance, the government could conduct ladder prices for household consumption of water to reduce water waste. Enterprises could apply water-saving devices and recycle water. Residents

could develop water saving habits, such as keeping the water for washing vegetables to flush the toilet.

Third, this study reveals the dual roles of environmental knowledge, namely both as a direct influential factor of water conservation and a moderator of the relationship between environmental concern and water conservation. This novel finding promotes the meaning and significance of this study. On the one hand, individuals with high environmental knowledge are prone to engage in water conservation. Environmental knowledge received through the mass media, such as watching documentaries or TV programs related to environmental protection, is beneficial in generating more water savers. On the other hand, environmental knowledge could also amplify the positive impact that environmental concern has on water conservation. In other words, environmental knowledge plays a catalytic role in the transformation of environmental concern into water conservation behavior. Based on the above discussion, this study suggests that the government could attach importance to water conservation education and increasing the public's household water conservation knowledge. For instance, keeping the water for washing vegetables to flush the toilet, turning off the faucet when brushing teeth, making sure the water faucets do not drip, and taking shorter showers are convenient methods for conserving water with low cost. Moreover, this study also suggests that the government could emphasize sustainability achievements and benefits of water conservation during environmental publicity.

Fourth, regarding the relationships between demographic variables and water conservation behavior, existing studies show that women are more likely to conserve water [64]. Aprile and Fiorillo [6] and Gilg and Barr [55] reveal that elder people are more likely to save water at home. Different from these findings of previous studies, this study finds that there is no significant difference in water conservation behavior among different genders and ages. Moreover, income exerts a significantly negative effect on water conservation behavior, consistent with the views of Aprile and Fiorillo [6], and Gilg and Barr [55]. A possible explanation behind this finding is that low-income residents have stronger economic motives to save water. Thus, the information about the demographic characteristics of water savers may shed light on more targeted water conservation policy design and managerial measures.

## 6. Limitations and Future Research

This study has several limitations that could be addressed in future research. First, the analysis is based on responses to self-reported measures, which are vulnerable to social desirability bias and therefore jeopardize the results' accuracy. Second, the research sample fails to cover six provinces and consists of mainly young residents. Consequently, the results may fail to represent the whole of China's population and subsequently also may not reflect the water conservation behaviors of old people. Therefore, future efforts are encouraged to address the limitation by investigating a more representative sample and to observe residents' actual water consumption behaviors. Additionally, future studies are also recommended to introduce more potential variables to increase the explaining power of the research model and build the analysis on applicable theories (e.g., Theory of Planned Behavior).

**Author Contributions:** Y.L., conceptualization, formal analysis, and writing—original draft; B.W., data curation, writing—review and editing, and methodology; M.C., conceptualization and critically revised the manuscript. All authors have read and agreed to the published version of the manuscript.

**Funding:** This research received no external funding.

**Institutional Review Board Statement:** Not applicable.

**Informed Consent Statement:** Informed consent was obtained from all subjects involved in the study.

**Data Availability Statement:** The data that support the findings of this study is available upon request to the corresponding author.

**Conflicts of Interest:** The authors declare no conflict of interest.

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
