# Peer review of "Environmental Concern, Environmental Knowledge, and Residents’ Water Conservation Behavior: Evidence from China"

_water, doi:10.3390/w14132087_

Round 1
Reviewer 1 Report
The study examines the effects of environmental concern and environmental knowledge on residents’ water conservation behavior. The findings indicate that residents with stronger environmental concerns and a higher level of environmental knowledge are more likely to exhibit water conservation in daily life in China. Additionally, environmental knowledge plays a moderating role in the relationship between environmental concern and water conservation behavior. Remarks: The abstract doesn’t show the accurate content and the main findings of the study area. Please add the main findings of the research work. What is the added value of this new study? Clearly define the objective of the work in the introduction. Please add more discussion material to the Results and Discussion section. What were perhaps different results from other studies, and why? The conclusion should be specific. It is recommended also to highlight the novelty and key findings of the work.
Author Response
Please see the attached file for the response details. Thank you for your time and the constructive suggestions.

Reviewer 2 Report
Water conservation is an important issue for creating a sustainable society. However, this study has many problems, and it has not reached the quality to be published in Water. Major issues are the unclear definitions of the concepts considered in this study and the insufficient review of the previous literature.
The authors defined water conservation as “water reuse” but this is not a common definition of water conservation. Water conservation can mean not only reusing water but also reducing water consumption. Then, the authors need to address the compatibility of their concept of water conservation and that of previous studies. Besides, it is not clear what “water reuse” meant for the survey participants.
The definitions of "environmental concern" and "environmental knowledge" have problems, too. For example, the explanatory variable that was used for measuring the level of “environmental knowledge” as a matter of fact measured the level of the propensity to obtain environmental knowledge via mass media. These two concepts are different from each other. There are established psychological scales of environmental concern, and the authors should explain why they chose the question among several candidate scales.
The authors discussed factors of environmental conservation behavior by referring to previous studies. Then, the major theories that relate the factors like environmental attitudes, behavioral intentions, and behavior need to be considered. The Theory of Planned Behavior by Icek Ajzen is one of them.
The study lacks the definition of the population that was targeted in their survey. This is a critical problem for a scientific report. Is it the whole individuals living in China? Is it the users of a specific survey application in China? Are there any differences in the composition of the sample from the targeted geographical population?
Although the authors concluded that environmental knowledge mediated the effects of environmental concerns on water reuse, the statistical results are indecisive. It is equally possible to interpret the same statistical results to mean that environmental concerns have mediated the effects of environmental knowledge on water reuse.
Author Response
Please see the attached file for response details. Thank you for your time and the constructive suggestions.
Best regards

Reviewer 3 Report
Please see attachment

Author Response
Please find the attachment for details of the response. Thank you for your time and the constructive suggestions.
Best regards

Round 2
Reviewer 1 Report
The study examines the effects of environmental concern and environmental knowledge on residents’ water conservation behavior. The findings indicate that residents with stronger environmental concerns and a higher level of environmental knowledge are more likely to exhibit water conservation in daily life in China. Additionally, environmental knowledge plays a moderating role in the relationship between environmental concern and water conservation behavior. Remarks: Line 125: The choice of reference could be supplemented with respect to the individuals who have insufficient environmental knowledge and unfamiliar with the approaches of environmental protection are unlikely to engage in green actions, even though they have high levels of environmental concerns.water management but also to emphasise their sustainability achievements when informing their customers, moreover, there need to be more instances of social participation, so that the different relevant sectors – government, private and civil – can all be involved in decision making and the development of strategic planning in [e.g. Ref [Consumers’ Perceptions of the Supply of Tap Water in Crisis Situations. Energies 2020, 13, 3617. https://doi.org/10.3390/en13143617. Add some information about the perspective of the future work.
Author Response
Thank you for your time and efforts. We appreciate it. Please see the attached file for more response details.
Best

Reviewer 2 Report
Although the authors have made efforts to improve the accuracy and academic contributions of the article, there are still important issues to be addressed.
The most important issue is the inability to test Hypothesis 3. As I have noted in my previous comment, this statistical model cannot distinguish whether the environmental knowledge mediates environmental concern or environmental concern mediates environmental knowledge. The authors admitted this problem in their response letter to me. The evidence that supports the notion that “environmental knowledge plays a positively moderating role in the relationship between environmental concern and water conservation behavior” is the knowledge provided by previous researchers and the authors’ own results have not played a decisive role. This issue needs to be clearly addressed for academic integrity. I think the following flow of logic may be acceptable.
1. Hypothesis 3 is rephrased to mean either or both of the above relationships exist.
2. Test this hypothesis and obtain an affirmative result
3. Discuss which direction of the relationships are more plausible by referencing to previous studies.
The authors should distinguish the types of water conservation activities between their own study and previous studies. We need to be careful about this difference when comparing the factors of “water reuse” and other types of water conservation. Some types of activities seem to be easier to do than “water reuse.” Then relevant factors may change.
Further discussions are needed to justify the selection of the indexes and questions for this study.
- Why did the authors choose environmental knowledge and environmental concern among potential factors of environmental conservation?
- Why did they choose the two questions among the original 16 questions by Minton and Rose (1997)? Before examining the reliability of the two selected questions, the validity of selecting these two questions out of 16 needs to be established.
The issue of the survey population has not been addressed. The authors should at least show how the distributions of the personal attributes among the surveyed population differ from the regional statistics for the 25 provinces. The authors are encouraged to discuss how the results of this study could be once a random sample from the geographical area has been taken and analyzed.
Author Response

(The authors gave the same response as above.)

Reviewer 3 Report
The authors complied with the comments and significantly improved the quality of the manuscript. Minor statements include extracting section 6 (line 309) and bringing the literature list in line with Water's journal guidelines (still not correct)
Author Response

(The authors gave the same response as above.)

Round 3
Reviewer 2 Report
The authors have addressed my previous concerns and nicely edited their article.
To further clarify the method for collecting respondents, I would like to ask the authors to specify the names of the social networking services that they used.
Author Response
Thank you for your time and efforts. We really appreciate it. Please find the attachment for more response details.
Best
